# HELLoRA: Hot Experts Layer-level Low-Rank Adaptation for MOE Model

## Abstract

Low-Rank Adaptation (LoRA) has become the dominant paradigm for Parameter-Efficient Fine-Tuning (PEFT) of large language models. However, most prior work focuses on dense architectures. In contrast, Mixture-of-Experts (MoE) models—now a de facto standard—scale parameter counts while keeping per-token compute nearly constant, creating new challenges for LoRA: how to minimize trainable parameters and maximize fine-tuning throughput without sacrificing quality. We propose Hot-Experts Layer-level Low-Rank Adaptation (**HELLoRA**), a simple yet effective scheme that attaches LoRA modules only to the hot experts at each layer, i.e., those most frequently activated. This design sharply reduces the number of trainable parameters and boosts fine-tuning throughput, while, perhaps unexpectedly, improving downstream performance. To stress-test HELLoRA under extreme parameter budgets, we further introduce **HELLoRI**, an orthogonal composition of HELLoRA with the recent LoRA with Reduced Interference (LoRI). Across extensive experiments on code generation, mathematical reasoning, and safety alignment, HELLoRA consistently outperforms strong PEFT baselines. In particular, relative to vanilla LoRA, HELLoRA uses only **15.74%** of the model parameters, improves accuracy by **9.24%**, and achieves an **88.80%** speedup. HELLoRI matches LoRA's accuracy while training just **0.7%** of LoRA's parameters. These results suggest that focusing LoRA capacity on hot experts is a practical path to scaling PEFT for large MoE LLMs.

## 1 Introduction

Mixture-of-Experts (MoE) models expand parameter capacity while keeping per-token compute nearly constant (Shazeer et al., 2017; Yun et al., 2024). They surpass dense large language models (LLMs) under matched compute and show strong performance across domains (Liu et al., 2024a; Jiang et al., 2024; Comanici et al., 2025; Muennighoff et al., 2025). Despite this efficiency, deployment remains costly because MoE models carry a large number of expert parameters. The cost is higher when task-specific fine-tuning is required. To curb resource usage, researchers have developed parameter-efficient fine-tuning (PEFT) methods (Hu et al., 2022). Among them, LoRA (Hu et al., 2022) is widely adopted for its balance between quality and efficiency. However, most LoRA variants () target dense architectures. Applying them directly to large MoE models introduces substantial memory overhead and stresses training throughput and hardware resources. Modern MoE models also exhibit sparse activation, where only a small subset of experts is active at each step, together with pronounced domain specialization (Muennighoff et al., 2025). These characteristics call for LoRA designs that are tailored to the MoE setting.

Inspired by the sparse activation and domain specialization of MoE models, a natural idea is to place the LoRA matrices $A$ and $B$ only on experts that are frequently activated. The first challenge is to identify these hot experts. During pretraining, a load-balancing loss $\mathcal{L}_{LB}$ (Shazeer et al., 2017) such as Eq. 1 is commonly applied to improve efficiency and generalization.

$$\mathcal{L}_{LB} = N_E \cdot \Sigma_{i=1}^{N_E} f_i \cdot P_i \tag{1}$$

where $N_E$ denotes the number of experts, $f_i$ is the fraction of tokens allocated to expert $Ei$, and $P_i$ is the corresponding routing probability.

Although some formulations of the MoE load-balancing loss apply it independently to each MoE layer, the widely used implementations, such as OLMoE and Mixtral, compute a single auxiliary

loss over concatenated router logits from all MoE layers, effectively balancing experts with the same index across layers (for example, expert 1 in layer 0 and expert 1 in layer 10). Under this implementation, we observe that global index-wise usage is approximately balanced (Fig. 1, orange), yet within individual layers and downstream tasks, expert activations can still be strongly imbalanced. Prior work (Muennighoff et al., 2025) and our measurements both reveal strong intra-layer domain specialization. For instance, in layer 7 with 64 experts, the eight most frequent experts account for more than 50% of activations (Fig. 1 green line).

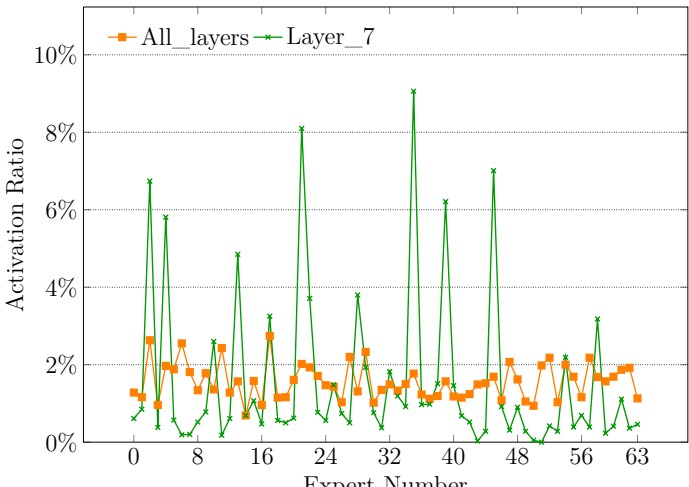

Figure 1: Expert activation ratio in OlMoE on the GSM8K dataset.

These observations motivate **Hot Experts Layer-level Low-Rank Adaptation (HELLoRA)**. HELLoRA assigns LoRA only to the top-$k$ most frequently activated experts in each layer. This reduces the adapter parameters in MoE layers to a fraction $k/n$ of dense LoRA, where $n$ is the number of experts. Other linear components, such as attention and gating, follow the standard LoRA recipe.

Recent work (Yu et al., 2024; Panda et al.) shows that updates from standard LoRA contain substantial redundancy. Independent studies (Zhang et al., 2023; Zhu et al., 2024) also find that freezing the up-projection $A$ can match training $A$ in accuracy. Pushing this idea further, LoRI (Zhang et al., 2025) reports that freezing $A$ and updating only 10% of the down-projection $B$ still yields strong performance. To examine HELLoRA under extreme parameter budgets, we propose HELLoRI. HELLoRI adopts the LoRI principle within our hot-expert scheme by freezing $A$ during fine-tuning and updating only 10% of $B$.

To evaluate HELLoRA and HELLoRI, we conduct extensive experiments on benchmarks spanning mathematical reasoning, code generation, and safety alignment. Using the OlMoE-1B–7B MoE models as base models, HELLoRA consistently outperforms LoRA and other PEFT baselines in accuracy. It cuts trainable parameters by up to **85%** relative to LoRA and increases training throughput by nearly **2x**. A key difference from masking schemes such as LoRI is that HELLoRA is the first to both reduce parameters and raise throughput. Masking methods can freeze the $A$ matrix to save gradient memory, yet $A$ still occupies memory and is computed in the forward pass. They also sparsify the $B$ matrix by multiplying it with a mask, which forces many entries to zero but does not reduce FLOPs or memory traffic and can introduce small extra overheads. In parallel, HELLoRI reaches performance comparable to LoRA and other PEFT approaches while training about **0.7%** of LoRA's parameters.

## 2 BACKGROUND AND MOTIVATION

### 2.1 LoRA

Pretrained large language models (LLMs) typically require task- or domain-specific adaptation to fully realize their expressive capacity. However, early full-parameter finetuning and more recent

reinforcement-learning–based post-training regimes demand prohibitive compute and memory when model sizes are large, making them infeasible for resource-constrained labs and small companies. This motivates parameter-efficient approaches that achieve competitive in-domain gains while updating only a small fraction of weights.

Low-rank adaptation (LoRA) was proposed against this backdrop Hu et al. (2022). Its rationale traces to linear-algebraic matrix factorization: as in 2, a large, not-full-rank (effectively sparse) matrix can be expressed as the product of two smaller matrices.

$$M_{n \times n} = A_{n \times k} \times B_{k \times n} \tag{2}$$

Leveraging the empirical low-intrinsic-dimension of neural updates, LoRA injects trainable low-rank factors into linear layers so that, rather than updating a full $\Delta \in \mathbb{R}^{d_{in} \times d_{out}}$ weight, optimization proceeds on two much smaller matrices $A \in \mathbb{R}^{d_{in} \times k}$ and $B \in \mathbb{R}^{k \times d_{out}}$ with $k \ll d_{in}, d_{out}$, as formalized in equation 3.

$$h = x \times (W + \Delta) = xW + xAB \tag{3}$$

where $W$ is the pretrained weight parameter, $\Delta$ is the LoRA adapter.

LoRA can be applied to standard linear components, e.g., attention projections $W_v, W_k$ and MLP projections $W_{up}, W_{down}$. A large body of subsequent work refines the structure, placement, and optimization of the low-rank factors to improve usability and effectiveness across tasks Liu et al. (2024b); Meng et al. (2024); Wu et al. (2024).

A complementary line of research aims to minimize trainable parameters while preserving accuracy. Recent studies Yu et al. (2024); Panda et al. observe that many parameters exhibit negligible change before vs. after LoRA finetuning; adapting those locations wastes compute and memory. Accordingly, several methods freeze $\boldsymbol{A}$ entirely with minimal accuracy loss, roughly halving the number of updated parameters (Zhang et al., 2023; Zhu et al., 2024). Pushing further, LoRI (Zhang et al., 2025) freezes $\boldsymbol{A}$ and updates only a subset (e.g., 10%) of $\boldsymbol{B}$'s entries while still converging competitively on multiple datasets.

Despite their parameter efficiency, these approaches share two fundamental limitations. First, they reduce the **number** of trainable parameters but do not necessarily reduce **compute**. For instance, LoRI introduces an additional masking matrix $M$ so that, as in equation 4,

$$h = x \times (W + \Delta) = xW + xA(B \odot M) \tag{4}$$

where $\odot$ represents element-by-element multiplication. The forward still performs the same (or even more) multiply–accumulate operations as vanilla LoRA in equation 3, plus extra masking overhead. Second, most techniques target dense LLMs and overlook the **sparse-activation characteristics of modern MoE architectures**, which open a broader design space for compute- and memory-efficient adaptation in both prefill and decode.

## 2.2 MOTIVATION

Our core motivation stems from the sparse activation behavior of modern MoE LLMs, which creates additional headroom to further reduce LoRA updates. As illustrated in Fig. 1, we empirically quantify expert activations in MoE layers. Even with a load-balancing loss in equation 1, MoE models exhibit pronounced expert-load skew during finetuning: only a small subset of experts is activated frequently, and correspondingly only those experts' parameters are updated often. Combined with prior observations that LoRA updates concentrate on a small fraction of locations within each adapted weight Yu et al. (2024); Panda et al., this yields a double sparsity effect: most expert-layer parameters are rarely touched by meaningful updates.

This observation suggests a simple but powerful design principle: do not attach LoRA adapters to cold experts. By allocating LoRA only to frequently activated experts, we can eliminate a large fraction of LoRA compute and memory at the source, rather than merely freezing parameters while still paying forward/Backward costs. In short, MoE's activation sparsity provides a principled lever to reduce both the number of trainable parameters and the actual FLOPs/memory footprint of adaptation.

## 3 METHOD

Our approach mainly consists of Layerwise Hot-expert Catcher and Efficient Finetuning for MoE. As shown in Fig. 2, hot-experts are first obtained by using the classical LoRA method (Fig. 2(a)) on a sub-dataset. Then, we accomplish efficient parameter fine-tuning using the method shown in Fig. 2(b).

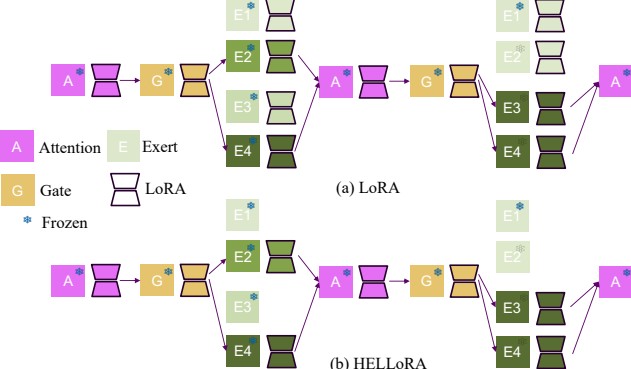

Figure 2: The execution process of LoRA and HELLoRA.

**Layerwise Hot-expert Catcher.** Prior work shows that hot experts vary across tasks for the same model Muennighoff et al. (2025). This reflects specialization across MoE layers. The mapping from tasks to experts is hard to know a priori. We therefore propose a layerwise hot-expert catcher. It runs a short, full-LoRA warm-up on a small sample of the target dataset. During this pass, it records the expert activation histogram for each MoE layer. It then selects the hot experts per layer and builds an expert-adapter list (the HELLoRA list). We finally attach adapters only to those experts and proceed to finetune on the full dataset.

**Efficient Finetuning for MoE.** HELLoRA follows standard LoRA practice. The base model is frozen. Only the adapter factors $A$ and $B$ are trainable. We write the low-rank update as $AB$ for brevity. Shapes follow the target linear. In each forward pass, a linear with an adapter fuses the base output with $AB$. The fused output is then passed to the next layer. Backpropagation updates only $A$ and $B$. Initialization follows common LoRA defaults. $A$ is initialized near zero to keep the function unchanged at step zero. $B$ uses a small random scale. A fixed scaling factor controls the update magnitude. Rank is a tunable knob. Higher rank increases capacity. Lower rank saves memory.

The key change is placement. We attach adapters only to experts in the HELLoRA list. We do not adapt every expert in an MoE layer. This reduces trainable parameters. It also reduces optimizer state and gradient traffic. The forward calls fewer adapter kernels. The backward builds fewer gradients. We keep the surrounding blocks intact. LayerNorm and residual paths are unchanged. Dropout and attention masks follow the base model. Mixed precision is used as in pretraining. No extra masks are introduced inside the adapter path.

We also explore adapters on attention projections and on the gating network. We test them alone and in combination in Section 5. Attention adapters target token transformations. Gate adapters target routing. The final placement balances accuracy and efficiency.

At inference time the adapters can be merged into the base weights or kept factorized. Merging removes extra matmuls. Keeping them factorized preserves modularity. Both options are supported without changing the training recipe.

**HELLoRI.** To push parameter count lower, we implement HELLoRI. HELLoRI freezes all $A$ factors. It updates only 10% of the entries in each $B$ matrix. The chosen entries follow LoRI's rule Zhang et al. (2025): select the top 10% by absolute value among all elements of $B$ at initialization or at the measured criterion. This preserves accuracy while cutting trainable parameters further.

# 4 EXPERIMENTS

## 4.1 EXPERIMENTAL SETUP

**Datasets.** We evaluate HELLoRA and HELLoRI across multiple domains. For each domain we finetune on a task-specific dataset and report test accuracy and end-to-end finetuning time.

1. Math reasoning. Finetune on GSM8K Cobbe et al. (2021) train and report accuracy on its test split.
2. Code generation. Finetune on CodeAlpaca Chaudhary (2023), then evaluate on HumanEval Chen et al. (2021) with pass1, pass5, and pass10.
3. Safety alignment. Finetune on Saferpaca Bianchi et al. (2023), then measure refusal rate on harmful queries from HEx-PHI Qi et al. (2024).

**Models and baselines.** All experiments use the state-of-the-art open MoE model OlMoE-1B-7B. It has ∼7B parameters in total, with ∼1B activated per step. The model contains 16 decoder layers and 64 experts per layer, with 8 experts activated at each step. We compare HELLoRA to full finetuning (FFT), vanilla LoRA, and two LoRI variants: LoRI-D (freeze $A$) and LoRI-S (LoRI-D plus updating only 10% of entries in $B$). Training uses FSDP Zhao et al. (2023) on two NVIDIA A100 GPUs. Hyperparameters, including learning rate, rank, and batch size, are listed in Table 1. All hyperparameters are derived from the optimal values obtained from the grid search.

Table 1: Hyperparameters in the Experiments

| Name | Value |
|---|---|
| Adapter Rank ($k$) | 32 |
| Batch size | 256 for FFT and 128 for others |
| Math Learning Rate | 4e-5 for FFT and 4e-4 for others |
| Code Learning Rate | 2e-4 for FFT and 2e-3 for others |
| Safety Learning Rate | 2e-5 for FFT and 4e-4 for others |
| Math Training Epoch Rate | 3 |
| Code Training Epoch | 2 |
| Safety Training Epoch | 1 |
| Samples for Catcher | 10% |
| Hot Expert Number | 8 |
| Scaling Factor $\alpha$ | 64 |
| Repeat Run Times | 10 with different seeds and show average |

We run the warm-up procedure and compute the Jaccard overlap of the warm-up stage selected hot experts per layer with real hot experts (new Fig. 3). We find that, across the 16 MoE layers, 11 layers have a Jaccard overlap of 1.0 (i.e., exactly the same hot-expert set is selected under all dataset), and in the remaining 5 layers the sets differ by only a single hot expert. The results show that the hot experts can be accurately identified using only 10% of the dataset.

## 4.2 LAYER-WISE EXPERT ACTIVATION PATTERNS

To better understand how MoE capacity is utilized in practice and to motivate the design of HEL-LoRA, we analyze the layer-wise expert activation patterns under different downstream tasks. Concretely, for each of our main tasks, we log the frequency with which each expert is selected at layers 0, 4, 8, 12, and 15, and visualize the resulting histograms in Fig. 4. We additionally report the same analysis on the Mixtral-8×7B model in Fig. 6.

Two consistent phenomena emerge from these plots. First, within a given MoE layer, the activation distribution is highly skewed: a small subset of experts are selected very frequently ("hot" experts), while a long tail of experts are rarely used ("cold" experts). This intra-layer sparsity holds across all considered tasks and for both OLMoE and Mixtral-8×7B, despite the presence of an auxiliary load-balancing loss in pre-training. Second, the identity of hot experts is clearly task-dependent: experts that are dominant for one task can become nearly inactive for another, and vice versa. In other words, expert importance is both *layer-specific* and *task-specific*.

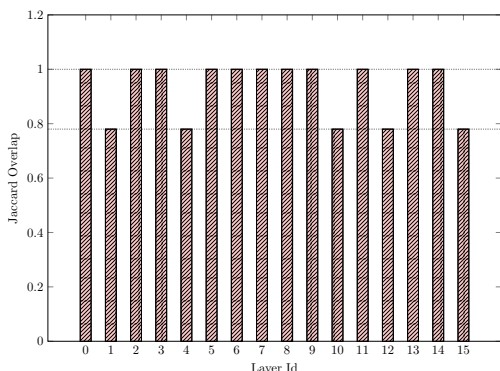

Figure 3: Jaccard Overlap of the Warm-up Stage Selected Hot Experts per Layer with Real Hot Experts.

These empirical observations directly motivate the HELLoRA design. Since only a small number of experts per layer carry most of the task-specific load, it is natural to focus adaptation capacity on these hot experts while freezing the cold ones. Moreover, the task-specific variation in the histograms suggests that hot experts should be selected *per layer and per task*, rather than using a single global subset. Our layerwise hot-expert catcher precisely operationalizes this idea by profiling the activation distribution on a small warm-up slice of the target data and then attaching LoRA modules only to the most frequently activated experts in each layer. As shown in Sec.4.4.3, this activation-aware selection enables HELLoRA to match or surpass full fine-tuning accuracy while updating only a small fraction of MoE parameters.

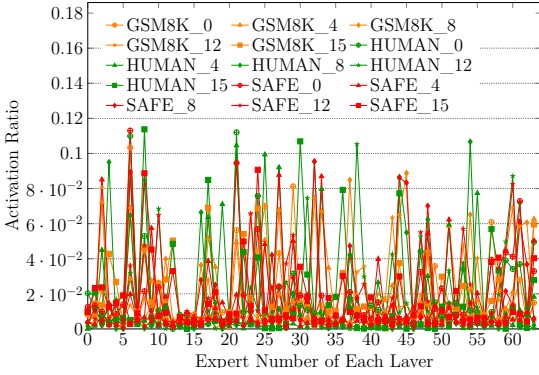

Figure 4: Expert activation ratio in $0, 2, 4, 8, 15$ layers in OlMoE on the three datasets.

### 4.3 ACCURACY RESULTS

Table 2 reports results on math, code, and safety benchmarks. FFT updates 100% of parameters. LoRA reduces the share of updated parameters to 4.3%. LoRI-D and LoRI-S push this to 1.9% and about 0.2%. HELLoRA, HELLoRI-D, and HELLoRI-S go further by shrinking expert-layer updates to 0.7%, 0.3%, and 0.03%, respectively.

Despite the tiny update budgets, HELLoRA delivers strong gains. It outperforms FFT, LoRA, and LoRI on math and code benchmarks. On HumanEval it leads on pass@1 and pass@5, with a small deficit to LoRA on pass@10 only. HELLoRI-D and HELLoRI-S match or exceed LoRI-D and LoRI-S across benchmarks. Unlike LoRI, which can occasionally surpass LoRA, HELLoRI does not overtake HELLoRA.

These findings suggest that, for MoE models, updating only hot experts is enough to reach excellent finetuning quality. The results also support the OlMoE observation of selective expert activation. Experts specialize for task types. Load-balancing losses do not erase this pattern. Adapting experts

Table 2: Accuracy performance comparison of different adaptation methods on GSM8K (math), HumanEval (code), and HEx-PHI (safety) benchmarks using OlMoE-1B-7B.

| Method | Params | GSM8K | HUMAN (@1/@5/@10) | SAFE |
|--------|--------|-------|-------------------|------|
| FFT | 100% (6.92B) | 27.64 | 16.4/21.04/22.38 | 97.32 |
| LoRA | 4.300% | 26.37 | 16.52/21.80/**24.03** | 98.75 |
| DoRA | 4.350% | 26.82 | 16.72/21.93/23.68 | 98.86 |
| LoRI-D | 1.900% | 28.22 | 16.98/21.85/23.75 | 96.56 |
| LoRI-S | 0.190% | 25.49 | 16.46/20.22/22.37 | 98.44 |
| HELLoRA | 0.700% | **29.49** | **17.87/22.04**/23.82 | **99.06** |
| HElLoRI-D | 0.300% | 28.42 | 16.89/20.89/22.21 | 98.75 |
| HElLoRI-S | 0.030% | 25.10 | 16.22/20.27/21.76 | 97.50 |

that are weak for the target task—or generic experts—may even hurt accuracy. HELLoRA's consistent edge over LoRA supports this view. Section 4.5.1 provides ablations on expert selection that reinforce this point. Restricting updates to a small subset of highly active experts acts as a form of structured regularization, preserving useful capabilities in less-active experts and reducing overfitting on the target distribution. This is consistent with prior observations on parameter-efficient finetuning and provides a plausible explanation for the observed gap. To further validate the generalizability of HELLoRA, we conducted additional performance comparisons between HELLoRA and other LoRA fine-tuning methods on Mixtral-8×7B. HELLoRA consistently achieves higher accuracy using fewer adapter parameters.

Table 3: Accuracy performance comparison on GSM8K benchmark using Mixtral-8×7B. The number of hot experts equals the number of experts activated per layer (i.e., $k = 2$).

| | LoRA | LoRI-D | LoRI-S | HELLoRA | HELLoRI-D | HELLoRI-S |
|--------|------|--------|--------|---------|-----------|-----------|
| **Params** | 1.03% | 0.59% | 0.06% | 0.31% | 0.17% | 0.02% |
| **Accuracy** | 67.73 | 64.84 | 63.98 | **68.28** | 64.61 | 66.17 |

We also note a broader implication. HELLoRA identifies hot experts on the pretrained base model. It therefore preserves task-relevant knowledge that is already embedded by pretraining. This aligns with the view that supervised finetuning mainly unlocks existing capabilities rather than creating new ones Liu et al. (2024b); Yu et al. (2024). Finally, sparsifying adapters further with HELLoRI acts as a strong regularizer but brings little accuracy gain over HELLoRA. A likely reason is that MoE already imposes layer-level sparsity, which provides regularization. In contrast, for already-sparse MoE adapters under LoRI, keeping only hot experts (HELLoRI) still cuts trainable parameters by a large margin without harming accuracy.

## 4.4 EFFICIENCY RESULTS

Accuracy usually increases with computational cost. HELLoRA targets only task-relevant parameters. It avoids adapters on cold experts and skips low-salience updates. This cut in redundant work improves accuracy and reduces step time.

We report end-to-end throughput in samples per second. The metric includes forward, backward, optimizer updates, and data movement. Hardware, precision, batch size, and optimizer are held fixed. Under this setting, Although we introduced approximately 16% additional overhead during the full fine-tuning hot expert determination phase on the 10% dataset, HELLoRA and HELLoRI reach about 1.62x the throughput of LoRA or LoRI, and about 3.5x that of FFT, as shown in Figure 5. The size of the warm-up dataset involves a trade-off between speed and expert selection accuracy, with potential for further exploration. The gain is consistent across math, code, and safety tasks.

The speedup comes from fewer adapted sites. Adapter FLOPs scale with the number of adapted linears. HELLoRA attaches adapters only to hot experts, which are a small subset of all experts. MoE layers then pay updates only when hot experts are active. Backward FLOPs and gradient volume drop in tandem. Optimizer state also shrinks. With fewer trainable parameters, Adam keeps fewer moments. Peak memory falls, which enables larger micro-batches at the same GPU budget. FSDP communication drops for the same reason.

Mask-based sparsification behaves differently. LoRI adds mask tensors and gating ops. These introduce extra reads and writes and limit kernel fusion. The forward still performs the base multiplies. The backward still builds gradients at masked sites. Net compute does not fall much, and the mask overhead remains. This explains why LoRI-S lags in throughput.

HELLoRI goes further on parameter count. It freezes $A$ and updates only 10% of $B$. It inherits the hot-expert placement from HELLoRA. In practice, HELLoRI matches or slightly exceeds HELLoRA in speed, but not in accuracy. The main driver of the Pareto gain is the expert-level selection. Extra sparsity inside B adds little once MoE layer sparsity is exploited. Overall, HELLoRA and HELLoRI reduce the number of MoE adapters to about one eighth of prior placements, yielding a real drop in trained parameters and a clear wall-clock gain.

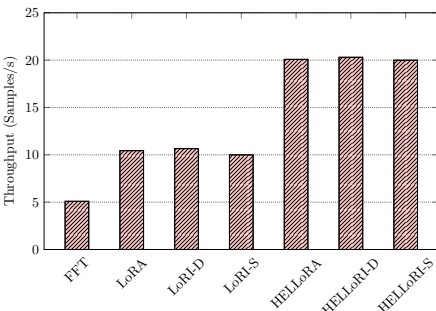

Figure 5: Training Throughput in OlMoE on the GSM8K dataset.

### 4.5 ABLATION STUDIES

#### 4.5.1 EXPERT SELECTION

We test whether the gains come from precise hot-expert targeting. We compare two alternatives. The first uses model-level hot experts. It counts cumulative activations across all layers and selects the top eight experts globally. The same eight adapters are then attached in every layer. The second uses cold-expert placement. It selects, per layer, the eight least activated experts and attaches adapters only to them.

We train on GSM8K and evaluate accuracy on its test split. HELLoRA and HELLoRI use the same ranks, batch sizes, learning rates, hardware, precision, and training length. Results appear in Table 4. Layerwise hot-expert selection clearly beats model-level selection. The gap holds for both HELLoRA and HELLoRI. This indicates that expert hotness is a layer-level property. Model-level aggregation mixes layers with different functions, and the load-balancing loss further smooths the global histogram. The global top-k set is therefore suboptimal for most layers.

Adapting hot experts also outperforms adapting an equal number of cold experts. Final accuracy is higher when adapters target hot experts. Cold-expert placement lags across settings. Expert hotness thus correlates with accuracy on GSM8K and supports the design choice behind HELLoRA: place adapters where the model already concentrates activation on the target task.

#### 4.5.2 LAYER SELECTION

We then study whether adapters are needed on attention and on gating layers in MoE models. We design two variants. _pure_ places adapters only on hot MoE experts. It omits adapters on attention and on the gate. _gate_ starts from _pure_ and adds adapters to the gate. Attention remains without adapters.

We evaluate with HELLoRA and with HELLoRI on math, code, and safety datasets. The training recipe is held fixed across variants. Table 5 reports accuracy. Removing adapters from attention and from the gate reduces the count of trainable parameters. It also produces a clear accuracy drop across domains.

Adding gate adapters on top of _pure_ recovers part of the loss. The gap to full HELLoRA placement remains. This indicates that attention adapters enable stronger task transfer. Gate adapters mainly

Table 4: Accuracy Performance comparison of different expert selection methods on GSM8K (math) benchmark using OlMoE-1B-7B. Bold indicates the best-performing method. _M represent model layer hot experts selection and _C means cold expert selection

| Method | Params | GSM8K |
|---|---|---|
| HELLoRA | 0.700% | **29.49** |
| HElLoRI-D | 0.300% | 28.42 |
| HElLoRI-S | 0.030% | 25.10 |
| HELLoRA_M | 0.700% | 28.51 |
| HElLoRI-D_M | 0.300% | 23.14 |
| HElLoRI-S_M | 0.030% | 23.14 |
| HELLoRA_C | 0.700% | 25.88 |
| HElLoRI-D_C | 0.300% | 25.19 |
| HElLoRI-S_C | 0.030% | 23.04 |

improve routing among experts. Attention adapters shape token transformations within and across experts. They adjust how information is mixed across heads and layers. Gate adapters refine which experts fire for a given token. Together they help, but attention provides the larger share of the gain.

Table 5: Accuracy performance comparison of different layer selection methods on GSM8K (math), HumanEval (code), and HEx-PHI (safety) benchmarks using OlMoE-1B-7B.

| Method | Params | GSM8K | HUMAN (@1/@5/@10) | SAFE |
|---|---|---|---|---|
| HELLoRA | 0.700% | **29.49** | **17.87/22.04/23.82** | **99.06** |
| HElLoRI-D | 0.300% | 28.42 | 16.89/20.89/22.21 | 98.75 |
| HElLoRI-S | 0.030% | 25.10 | 16.22/20.27/21.76 | 97.50 |
| HELLoRA_pure | 0.543% | 26.46 | 17.07/21.19/22.78 | 98.43 |
| HElLoRI-D_pure | 0.241% | 23.24 | 17.07/20.38/21.89 | 91.25 |
| HElLoRI-S_pure | 0.020% | 23.44 | 16.15/20.82/22.29 | 97.81 |
| HELLoRA_gate | 0.558% | 28.03 | 17.09/21.86/23.28 | 97.50 |
| HElLoRI-D_gate | 0.242% | 23.53 | 16.95/21.80/23.17 | 90.32 |
| HElLoRI-S_gate | 0.020% | 23.44 | 16.06/21.17/22.73 | 96.56 |

As shown in Table 6, we show the variation of _pure and _gate throughput under various algorithms. The efficiency gain from removing attention and gate adapters is small in wall clock. The memory saving is also modest once expert adapters are already sparse. The accuracy loss dominates the trade-off. We therefore recommend keeping adapters on both attention and gate when using HELLoRA. This placement offers a better accuracy to efficiency balance in practice.

Table 6: Throughput performance (samples/s) comparison of different adaptation methods and layer selections on GSM8K (math), benchmarks using OlMoE-1B-7B. **All** means use adapters to attention, gate and hot MoE layers.

| Method | _pure | _gate | All |
|---|---|---|---|
| LoRA | 10.01 | 11.97 | 10.44 |
| LoRI-D | 10.95 | 10.98 | 10.65 |
| LoRI-S | 10.77 | 10.91 | 10.00 |
| HELLoRA | **19.60** | 19.20 | 19.08 |
| HElLoRI-D | 18.41 | 19.03 | 19.11 |
| HElLoRI-S | 19.45 | 19.02 | 19.07 |

The ablations point to one principle. Place capacity where the model already works. Layerwise hot-expert selection beats model-level selection on GSM8K. The signal lives at the layer level. Global aggregation blurs it. Adapting cold experts hurts accuracy. Adapting hot experts helps. Adapters on attention and on the gate also matter. Dropping both reduces parameters but lowers accuracy across math, code, and safety. Adding gate adapters on top of _pure recovers part of the loss. The gap to full HELLoRA remains. Attention adapters drive task transfer. Gate adapters improve routing.

Throughput gains from removing these adapters are small, and memory savings are modest once expert adapters are sparse. The accuracy loss dominates.

We therefore recommend layerwise hot-expert placement plus attention and gate adapters. This configuration offers the best balance of accuracy, efficiency, and parameter economy in practice.

### 4.5.3 THE NUMBER OF HOT EXPERTS ($k$)

To understand the impact of the number of hot experts per layer, $k$, we conduct a dedicated ablation on the GSM8K task, varying $k \in \{2, 4, 8, 16\}$ (see Tab. 7). We observe a clear accuracy-efficiency trade-off. When $k = 2$, HELLoRA achieves the highest throughput and the smallest number of trainable parameters, but at the cost of a noticeable drop in accuracy. Increasing to $k = 4$ still yields high throughput and a compact parameter budget, while the accuracy degradation becomes much smaller. Setting $k = 8$ provides a robust sweet spot, it matches or slightly surpasses the best full fine-tuning baseline in accuracy, while retaining most of the efficiency gains. Further increasing to $k = 16$ brings only marginal accuracy improvements, but substantially reduces the throughput advantage. Overall, these results justify our default choice of $k = 8$ and offer practitioners a concrete guideline on how to trade off accuracy versus efficiency when deploying HELLoRA.

Table 7: Comparison of HELLoRA accuracy and speed for different $k$ values in the GSM8K benchmark using the OlMoE-1B-7B model. The bolded entry indicates the optimal method.

|  | $k = 2$ | $k = 4$ | $k = 8$ | $k = 16$ |
|---|---|---|---|---|
| **Params** | 0.27 | 0.41 | 0.68 | 1.21 |
| **Accuracy** | 20.99 | 20.98 | 20.08 | 16.49 |
| **Throughput** | 26.72 | 26.97 | 29.49 | 29.69 |

Although setting $k$ to the number of experts activated in the model is generally a good choice. In practice, we recommend choosing $k$ in a data- and model-aware manner rather than relying on a single fixed value. As a rule of thumb, for simpler or more homogeneous tasks where the expert activation histogram is highly skewed, a smaller $k$ (e.g., $k = 4$) often suffices to capture most task-specific capacity while maximizing speedup. For more complex or multi-domain tasks, moderately larger values (e.g., $k = 8$) are preferable to avoid underfitting. In settings with new private data or unfamiliar MoE architectures, we suggest running a small pilot sweep over $k$ (such as $k \in \{4, 8, 16\}$) on a subset of the training data and selecting the configuration that best matches the desired accuracy–efficiency trade-off. This procedure typically requires only a few additional runs, but provides robust, empirically grounded choices of $k$ for real-world deployments of HELLoRA.

## 5 CONCLUSION

This work introduces HELLoRA, a parameter-efficient fine-tuning method for MoE LLMs that allocates LoRA capacity only to the most frequently activated ("hot") experts at each layer. This simple mechanism substantially reduces trainable parameters and increases fine-tuning throughput while improving downstream quality. Across math reasoning, code generation, and safety alignment, HELLoRA consistently outperforms strong PEFT baselines; relative to vanilla LoRA, it uses 15.74% of the trainable parameters, delivers an 88.80% throughput gain, and yields a 9.24% accuracy improvement. To further tighten the parameter budget, we compose HELLoRA with LoRI to obtain HELLoRI, which fixes the up-projection as a random map and sparsifies the down-projection with task-specific masks, achieving LoRA-comparable accuracy while training only 0.7% of LoRA's parameters. These results indicate that concentrating adaptation on hot experts is an effective and practical route to scaling PEFT for large MoE models.

**Future Work.** We outline three directions. First, because hot experts vary across tasks, HELLoRA is a natural basis for multi-task adapter merging and for continual learning with reduced cross-task interference relative to vanilla LoRA. Second, the design is modality agnostic, and we plan to extend HELLoRA to MoE-based diffusion models and vision–language models to enable multimodal generation. Third, base-model inference cost remains the primary bottleneck even when adapter parameters are heavily reduced, so we will explore methods to lower base-model compute without compromising fine-tuning accuracy.

## 6 ETHICS STATEMENT

We have read and will adhere to the ICLR Code of Ethics. Our study does not involve human subjects, user-generated private data, or interventions. All datasets are publicly available under their original licenses. We follow license terms and cite sources. We do not collect or release any personal or sensitive information. Our method targets parameter-efficient finetuning of MoE LLMs and can be applied in many domains. We will release code and configuration files to support reproducibility and auditing. We report training settings, model sizes, and evaluation protocols. We do not believe our work increases privacy or security risks beyond existing adaptation methods. Our approach lowers compute and memory during finetuning, which may lessen environmental impact relative to full-parameter training. We are not aware of conflicts of interest or undisclosed sponsorship related to this work. We will comply with all legal and institutional requirements for data use and research integrity.

## 7 REPRODUCIBILITY STATEMENT

We aim for full reproducibility. The Experimental Setup section details datasets, model variants, training schedules, and all hyperparameters. We include an anonymous, downloadable code package in the supplementary materials that contains scripts to reproduce all tables and figures end-to-end, plus a README with commands and expected outputs. Upon acceptance, we will release a public repository with the same code and configuration, preserving the anonymous artifact's commit and file structure for continuity.

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

## A    USE OF LARGE LANGUAGE MODELS (LLMs).

We used an LLM only as a writing assistant for grammar correction, wording polish, and minor clarity edits. The LLM did not generate ideas, design experiments, analyze results, write technical content, produce code, or draft figures and tables. All methods, claims, and conclusions are authored and verified by us. We reviewed every suggestion and accepted or rejected edits at our discretion. The LLM was not trained or fine-tuned on our data and did not access any nonpublic datasets. This disclosure follows the ICLR policy on LLM usage and is included in the appendix outside the page limit.

## B    LAYER-WISE EXPERT ACTIVATION PATTERNS IN MIXTRAL

Similar to the OlMoE model, Mixtral-8×7B also exhibits distinct expert sparse activation characteristics across different layers. As shown in Figure 6, the distributions of the Top-2 experts differ across layers, further validating the rationale behind the HELLoRA motivation.

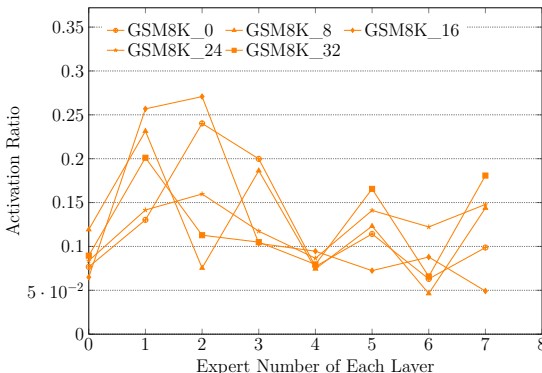

Figure 6: Expert activation ratio in $0, 8, 16, 24, 32$ layers in OlMoE on the three datasets.

## C  CROSS-TASK EVALUATION OF COLD EXPERT PRESERVATION

While Fig. 4 already shows that different downstream tasks activate distinct subsets of experts across layers, it remains unclear to what extent adapting only task-specific hot experts might harm capabilities that rely on other experts. To probe this question, we perform a cross-task evaluation that explicitly measures the impact of HELLoRA on tasks whose experts are mostly cold during fine-tuning.

Concretely, we consider three downstream tasks from our main benchmark and, for each task in turn, identify its layer-wise hot experts using the warm-up procedure described in Sec.4.1. We then fine-tune HELLoRA on this single target task by updating only its selected hot experts, keeping all other experts frozen as in our standard HELLoRA setting. After fine-tuning, we evaluate the resulting model not only on the target task, but also on the remaining two tasks, and compare their accuracies before and after fine-tuning. Table 8 summarizes the results.

Across all three target–auxiliary task combinations, we observe the following patterns. First, HEL-LoRA yields the expected improvements on the target task, confirming that restricting updates to task-specific hot experts does not hinder adaptation on that task. Second, for the non-target tasks, two of the auxiliary tasks actually show improved performance after fine-tuning, while the SAFE task exhibits only a slight degradation, and we do not observe any large or systematic degradation on auxiliary tasks. This holds despite the fact that, as shown in Fig. 4, the hot experts for different tasks have limited overlap, implying that many experts that are important for an auxiliary task remain effectively cold during fine-tuning on another task.

Table 8: Accuracy of Non-target Tasks Before (_B) and After (_A) Fine-tuning the Hot Experts on the Target task using OlMoE-1B-7B.

|         | GSM8K_B | GSM8K_A | HUMAN_B        | HUMAN_A        | SAFE_B | SAFE_A |
|---------|---------|---------|----------------|----------------|--------|--------|
| **GSM8K** | N/A   | N/A     | 13.41/16.81/18.17 | 13.78/18.94/20.63 | 66.75 | 58.43 |
| **HUMAN** | 1.66  | 5.86    | N/A            | N/A            | 66.75 | 63.01 |
| **SAFE**  | 1.66  | 2.83    | 13.41/16.81/18.17 | 15.18/19.51/21.33 | N/A   | N/A   |

These findings suggest that focusing updates on layer- and task-specific hot experts often preserves, and can occasionally even enhance, performance on other domains whose experts are left frozen. In other words, HELLoRA rarely exhibit strong catastrophic forgetting of capabilities associated with cold experts, alleviating the concern that expert sparsification would severely regress non-target domains. This cross-task behavior complements our single-task results and supports the use of HELLoRA in multi-task or sequential fine-tuning scenarios where retaining broad capabilities is important.

