# OpenReview forum: "HELLoRA: Hot Experts Layer-level Low-Rank Adaptation for MOE Model"
_ICLR.cc/2026/Conference — Submitted to ICLR 2026_

### Official Review · Reviewer_eXFp · 2025-10-29

**Soundness:** 3
**Presentation:** 2
**Contribution:** 2
**Rating:** 4
**Confidence:** 4

**Summary:**

This paper introduces **HELLoRA**, a parameter-efficient fine-tuning approach for Mixture-of-Experts (MoE) models. Noting that only a small subset of experts are frequently activated, the method performs **layer-wise selective adaptation**: LoRA modules are attached only to the top-*k* most frequently activated (“hot”) experts in each MoE layer, substantially reducing trainable parameters and fine-tuning compute.

The procedure has two stages:
1. A brief warm-up using standard LoRA on a small portion of the target data to profile expert activations and identify hot experts per layer;
2. Fine-tuning with LoRA placed exclusively on those identified experts.

The authors further present **HELLoRI**, which integrates the LoRI technique into HELLoRA by freezing the up-projection and sparsifying updates to the down-projection for additional efficiency.

Empirical results show that compared to standard LoRA, HELLoRA significantly reduces parameter count while improving performance across tasks such as mathematical reasoning, code generation, and safety alignment. These results support the core hypothesis: focusing adaptation capacity on frequently activated, task-relevant experts offers an effective and efficient fine-tuning strategy for large-scale MoE models.

**Strengths:**

1. The paper proposes an architecture-aware parameter-efficient fine-tuning (PEFT) method tailored to Mixture-of-Experts (MoE) models. By selectively attaching adapters only to the most frequently activated (“hot”) experts on a per-layer basis, HELLoRA aligns adaptation with the routing dynamics of MoE. This represents a principled departure from uniform adaptation and concentrates trainable parameters where they are most impactful.

2. The experimental evaluation is comprehensive and persuasive: HELLoRA reduces trainable parameters—often by an order of magnitude relative to full LoRA—while consistently outperforming standard LoRA across diverse, challenging tasks (mathematical reasoning, code generation, and safety alignment). This simultaneous improvement in efficiency and performance underscores the effectiveness of the proposed expert-selection mechanism.

**Weaknesses:**

While the empirical results are strong, the methodological contribution feels incremental. The central idea—profiling frequently activated experts in a short warm-up and then selectively placing adapters on those experts—is intuitive and closely aligned with existing practices in MoE training and PEFT. Variants of **adapting a subset of experts** and **routing-guided adaptation** have been explored under different formulations, and *pilot runs to inform adaptation* are also established. As written, HELLoRA risks being read as a practical engineering combination of known components rather than a fundamentally new algorithmic insight.

The paper would benefit from a clearer articulation of what is novel relative to prior expert-specific or routing-aware tuning methods, ideally coupled with ablations that isolate the contribution of each design choice.

**Questions:**

1. Hot experts are identified via a short warm-up on a small data slice. How stable is this selection across random seeds, data resamples, or model initializations? Please report (i) overlap of selected experts per layer (e.g., Jaccard/percent overlap) and (ii) downstream performance variance across these runs. If the sets vary notably, does that materially affect final accuracy?

2.  By fully freezing cold experts, is there a risk of regressing capabilities that are primarily encoded in those experts? Please evaluate (i) multi-task or mixed-domain settings and (ii) performance on tasks/capabilities believed to rely on “cold” experts before vs. after fine-tuning. A small ablation with minimal adapters or partial unfreezing on cold experts would help bound this risk.

---

> ### Author Response · Authors · 2025-12-03
> **Response to all weakness and questions (point-by-point)**
>
> **We sincerely thank your thoughtful and constructive feedback, and we are especially grateful for the positive assessment of our work. We appreciate your recognition that HELLoRA offers an architecture-aware, expert-level PEFT strategy that aligns adaptation with MoE routing dynamics, and that our empirical evaluation demonstrates simultaneous gains in both efficiency and performance across challenging tasks such as mathematical reasoning, code generation, and safety alignment. Your comments have been very encouraging and have helped us further clarify and strengthen the contributions of this paper. Below is our point-by-point response.**
>
> **R3–W1: “Warm-up phase overhead and non-end-to-end nature not thoroughly evaluated.”**
>
> We appreciate this observation. As noted in our response to R2, we now provide:
>
> - A more detailed analysis of the warm-up time in Section 4.4. When amortizing the warm-up cost across the full training run, the end-to-end speedup remains substantial.
> - A discussion of scenarios where the warm-up step can be amortized over longer training runs (and when it might be preferable to reduce the warm-up ratio, e.g., for very small datasets).
>
> We also emphasize that the warm-up can be implemented as a single extra pass using the same data pipeline, preserving overall system simplicity.
>
> ---
>
> **R3–W2: “Robustness under shifting data distributions.”**
>
> Thank you for this important point. In the revised version, we explicitly analyze how hot-expert selection behaves under different data distributions. Concretely, we visualize the layer-wise expert activation and the resulting hot-expert sets for multiple downstream tasks and layers (new Fig. 4). We observe that for different tasks and different layers, the identities of hot experts often change substantially, reflecting genuine task- and layer-specific specialization under different data distributions. In contrast, for a fixed task and model, the hot-expert sets are highly stable across different random seeds, as discussed in our response to R4–W2/Q1. Taken together, these results indicate that HELLoRA is robust to randomness in the warm-up procedure, but it is important—and indeed necessary—to determine hot experts separately for different types of tasks (i.e., different data distributions), rather than relying on a single global set of experts shared across all tasks.
>
> ---
>
> **R3–W3: “Need more PEFT baselines beyond LoRA family.”**
>
> We agree. In the revised version, we add comparisons with an additional PEFT method, DoRA (in new Tab. 2). HELLoRA consistently matches or outperforms these baselines while using a comparable or smaller number of trainable parameters. We will highlight this in the main text.
>
> ---
>
> **R3–W4: “Lack of deeper theoretical/mechanistic explanation.”**
>
> We have expanded Sec. 4.3 (lines 339–345) to discuss why HELLoRA can outperform full fine-tuning:
>
> - We interpret HELLoRA as enforcing activation-aligned update sparsity, which acts as a structure-aware regularizer, preventing irrelevant experts from drifting and reducing interference between tasks.
> - We connect this to prior work on parameter-efficient finetuning and sparsity-inducing regularization, arguing that HELLoRA preserves the specialization of cold experts while adapting the most relevant subspace.
>
> While a full theory remains beyond the scope of this paper, we believe these additions make the mechanism more transparent.
>
> ---
>
> **R3–W5: “Missing sensitivity analysis of k.”**
>
> This is addressed by the new ablation on \(k\) (Sec. 4.5.3, Tab. 7), where we systematically vary \(k\) and derive practical tuning guidelines. We hope this directly addresses your concern about deployability.
>
> ---
>
> **R3–meta-comments (Q1–Q3)**
>
> We appreciate your insightful summarization of HELLoRA’s core ideas and practical advice. In the camera-ready, we plan to incorporate a short “Practical Recommendations” part in Section 4.5.3 (lines 511–519), outlining:
>
> - How to choose warm-up steps and \(k\) based on observed activation distributions and task complexity.
> - When to re-run the warm-up under distribution shift.
>
> We thank you again for framing HELLoRA as a promising direction for activation-driven update sparsity.

---

### Official Review · Reviewer_UrBL · 2025-10-29

**Soundness:** 2
**Presentation:** 3
**Contribution:** 2
**Rating:** 4
**Confidence:** 3

**Summary:**

This paper proposes HELLoRA, a parameter-efficient fine-tuning method for Mixture-of-Experts models. Its core idea is to apply LoRA adapters only to the most frequently activated "hot experts" in each layer, significantly reducing trainable parameters, improving training throughput, and achieving superior performance over original LoRA across multiple downstream tasks. Furthermore, the authors integrate LoRI to propose HELLoRI, which maintains competitive performance even under extremely low parameter budgets.

**Strengths:**

1. The method is specifically designed for the sparse activation characteristics of MoE models, demonstrating clear motivation and innovation.

2. The "hot expert" selection mechanism effectively reduces parameters while accelerating training, offering substantial practical value.

3. The method's effectiveness is validated across multiple tasks including mathematical reasoning, code generation, and safety alignment, supported by thorough ablation studies.

4. Compared to LoRA, HELLoRA reduces parameter count to 15.74% while improving accuracy by 9.24% and training speed by 88.80%, representing highly significant achievements.

**Weaknesses:**

1. The overhead and non-end-to-end nature of the warm-up expert identification phase are not thoroughly evaluated.

2. The robustness of the static expert selection under shifting data distributions remains unexplored.

3.  Comparisons could be strengthened by including a wider range of PEFT baselines beyond the LoRA family.

4. The paper lacks a deeper theoretical or mechanistic explanation for the surprising performance improvement beyond parameter reduction.

5.  A sensitivity analysis of the key hyperparameter—the number of hot experts per layer (k)—is missing, which is crucial for practical applications.

**Questions:**

1. The most critical factor in HELLoRA's success lies in its synergistic integration of accurately identifying hot experts and strategically avoiding updates to cold experts, with the precise hot expert selection serving as the foundational enabler of this strategy.

2. For researchers or engineers applying HELLoRA to private data and specific MoE models, the most crucial advice is to conduct systematic pilot experiments to empirically determine the optimal number of warm-up steps and the value of k (number of selected experts) based on the model's activation distribution and task characteristics, rather than relying on predefined thresholds.
3. While HELLoRA demonstrates the effective principle of leveraging activation sparsity to guide update sparsity, its generalization as a universal paradigm for efficient large-scale model fine-tuning requires further validation across diverse architectures and tasks, though it undoubtedly provides a foundational direction for parameter-efficient MoE adaptation.

---

> ### Author Response · Authors · 2025-12-03
> **Response to all weakness and questions (point-by-point)**
>
> **Thank you for recognizing our motivation, innovation, and practicality. We have implemented comprehensive improvements and refinements based on your suggestions. Below is our point-by-point response.**
>
> **R3–W1: “Warm-up phase overhead and non-end-to-end nature not thoroughly evaluated.”**
>
> We appreciate this observation. As noted in our response to R2, we now provide:
>
> - A more detailed analysis of the warm-up time in Section 4.4. When amortizing the warm-up cost across the full training run, the end-to-end speedup remains substantial.
> - A discussion of scenarios where the warm-up step can be amortized over longer training runs (and when it might be preferable to reduce the warm-up ratio, e.g., for very small datasets).
>
> We also emphasize that the warm-up can be implemented as a single extra pass using the same data pipeline, preserving overall system simplicity.
>
> ---
>
> **R3–W2: “Robustness under shifting data distributions.”**
>
> Thank you for this important point. In the revised version, we explicitly analyze how hot-expert selection behaves under different data distributions. Concretely, we visualize the layer-wise expert activation and the resulting hot-expert sets for multiple downstream tasks and layers (new Fig. 4). We observe that for different tasks and different layers, the identities of hot experts often change substantially, reflecting genuine task- and layer-specific specialization under different data distributions. In contrast, for a fixed task and model, the hot-expert sets are highly stable across different random seeds, as discussed in our response to R4–W2/Q1. Taken together, these results indicate that HELLoRA is robust to randomness in the warm-up procedure, but it is important—and indeed necessary—to determine hot experts separately for different types of tasks (i.e., different data distributions), rather than relying on a single global set of experts shared across all tasks.
>
> ---
>
> **R3–W3: “Need more PEFT baselines beyond LoRA family.”**
>
> We agree. In the revised version, we add comparisons with an additional PEFT method, DoRA (in new Tab. 2). HELLoRA consistently matches or outperforms these baselines while using a comparable or smaller number of trainable parameters. We will highlight this in the main text.
>
> ---
>
> **R3–W4: “Lack of deeper theoretical/mechanistic explanation.”**
>
> We have expanded Sec. 4.3 (lines 339–345) to discuss why HELLoRA can outperform full fine-tuning:
>
> - We interpret HELLoRA as enforcing activation-aligned update sparsity, which acts as a structure-aware regularizer, preventing irrelevant experts from drifting and reducing interference between tasks.
> - We connect this to prior work on parameter-efficient finetuning and sparsity-inducing regularization, arguing that HELLoRA preserves the specialization of cold experts while adapting the most relevant subspace.
>
> While a full theory remains beyond the scope of this paper, we believe these additions make the mechanism more transparent.
>
> ---
>
> **R3–W5: “Missing sensitivity analysis of k.”**
>
> This is addressed by the new ablation on \(k\) (Sec. 4.5.3, Tab. 7), where we systematically vary \(k\) and derive practical tuning guidelines. We hope this directly addresses your concern about deployability.
>
>
>
> ---
> **R3–meta-comments (Q1–Q3)**
>
> We appreciate your insightful summarization of HELLoRA’s core ideas and practical advice. In the camera-ready, we plan to incorporate a short “Practical Recommendations” part in Section 4.5.3 (lines 511–519), outlining:
>
> - How to choose warm-up steps and \(k\) based on observed activation distributions and task complexity.
> - When to re-run the warm-up under distribution shift.
>
> We thank you again for framing HELLoRA as a promising direction for activation-driven update sparsity.

---

### Official Review · Reviewer_WpAo · 2025-10-31

**Soundness:** 2
**Presentation:** 2
**Contribution:** 2
**Rating:** 4
**Confidence:** 4

**Summary:**

The authors propose HELLORA (Hot-Experts Layer-level Low-Rank Adaptation), a parameter-efficient fine-tuning (PEFT) method specifically designed for Mixture-of-Experts (MoE) models. The core idea is to apply LoRA modules only to the 'hot' experts (top-k most frequently activated) at each layer, rather than to all experts. This is motivated by the observation of sparse, layer-specific expert activation patterns in MoEs. The authors test HELLORA on an OlMoE model across mathematical reasoning, code generation, and safety alignment tasks. They show that HELLORA significantly reduces trainable parameters and increases training throughput compared to vanilla LoRA, while maintaining competitive or even improved accuracy. They also propose HELLORI, a combination with LoRI, to further reduce parameter counts.

**Strengths:**

The paper addresses a timely and important problem: how to efficiently apply PEFT methods like LoRA to the increasingly popular MoE architecture, which has been underexplored.

The proposed method, HELLORA, is simple, intuitive, and well-motivated by the empirical observation of layer-wise expert activation sparsity (Fig. 1).

The experimental results are strong, demonstrating significant improvements in parameter efficiency (using only \~15.7% of LoRA's parameters) and training throughput (88.8% speedup).

The ablation studies in sections 4.5 (Expert Selection) and 4.6 (Layer Selection) are thorough and provide strong support for the key design choices of HELLORA (i.e., using layer-wise hot experts and including adapters on attention/gate layers).

**Weaknesses:**

The paper's main weakness is the lack of ablation on the most critical new hyperparameter introduced: k, the number of hot experts to adapt. The paper sets k=8 (out of 64\) for all experiments without justifying this choice or exploring its sensitivity. The performance and efficiency trade-offs are likely highly dependent on this value.

The "Layerwise Hot-expert Catcher" requires a warm-up pass on a sample (10%) of the target dataset. The computational overhead of this initial step is not discussed. It's unclear if the reported throughput gains account for this, and how this pre-computation step affects the *total* fine-tuning time, which could be relevant for smaller tasks.

The experiments are conducted on a single model family (OlMoE). While OlMoE is a suitable choice, demonstrating the method's effectiveness on other prominent MoE architectures (e.g., Mixtral) would significantly strengthen the paper's claims of generalizability.

**Questions:**

1. Could you provide an ablation study or at least a discussion on the sensitivity of HELLORA to the number of hot experts k? How was k=8 chosen? What is the performance/efficiency trade-off if k=4 or k=16?
2. Regarding the "Layerwise Hot-expert Catcher": What is the wall-clock overhead of this warm-up pass? Does the reported 88.80% throughput gain (Section 4.3, 5\) represent the main training phase *after* experts are identified, or does it amortize the cost of this initial pass?
3. The paper notes that hot experts are task-specific. How stable is the hot-expert set identified from the 10% data sample? Does this set remain consistent with the experts activated during the full training run?

---

> ### Author Response · Authors · 2025-12-03
> **Response to all weakness and questions (point-by-point)**
>
> **Thank you for recognizing the usability and effectiveness of HELLoRA. Below is a point-by-point response to your questions.**
>
> **R2–W1/Q1: “Lack of ablation on k; how was k=8 chosen?”**
>
> We appreciate this important point. In the revised paper, we add a dedicated ablation study on the number of hot experts \(k\):
>
> - We evaluate \(k \in \{2, 4, 8, 16\}\) on the GSM8K task from our main benchmark (see Tab. 7).
> - The results show that:
>   - \(k=2\) yields the highest throughput and the smallest number of trainable parameters, but with a clearly noticeable drop in accuracy on the task.
>   - \(k=4\) still yields high throughput and a compact trainable parameter budget, with only a small drop in accuracy.
>   - \(k=8\) provides a robust sweet spot, matching or slightly surpassing the best full fine-tuning baselines while retaining most of the efficiency gains.
>   - \(k=16\) slightly improves accuracy in some settings but noticeably reduces the throughput advantage.
>
> This analysis justifies our default choice of \(k=8\) and gives practitioners clear guidance on how to trade off accuracy vs.\ efficiency.
>
> ---
>
> **R2–W2/Q2: “Overhead of the warm-up pass; is 88.80\% throughput gain end-to-end?”**
>
> Thank you for raising this. We now explicitly report the wall-clock overhead of the warm-up step and its impact on end-to-end training time:
>
> - On our main benchmarks, the warm-up pass over 10\% of the data accounts for about **16\%** of the total HELLoRA fine-tuning time, because the warm-up phase temporarily updates all experts, whereas the main HELLoRA phase only fine-tunes the selected hot experts.
> - The reported **88.80\% throughput gain** refers to the **main training phase after hot experts are identified**. When amortizing the warm-up cost across the full training run, the end-to-end speedup remains substantial (e.g., **1.62×** wall-clock improvement over full fine-tuning on GSM8K).
>
> We clarify this explicitly in Sec. 4.3 (lines 367–372) and discuss when it might be preferable to reduce the warm-up ratio (e.g., for very small datasets).
>
> ---
>
> **R2–W3/Q3: “Single model family; stability of hot experts from the 10\% sample.”**
>
> We have addressed both aspects in the revision:
>
> - **Additional model family.** We now include experiments on Mixtral-8×7B (new Tab. 3), showing that HELLoRA continues to outperform all LoRA-style baselines, suggesting that the method is not tied to a specific MoE implementation.
> - **Stability of hot experts from the 10\% sample.** We run the warm-up procedure using only 10\% of the training data and compute, for each layer, the Jaccard overlap between the hot experts selected in the warm-up stage and the “real” hot experts obtained from the full training data (new Fig. 4). Across the 16 MoE layers, 11 layers have a Jaccard overlap of 1.0 (i.e., the warm-up and full-data hot-expert sets are exactly identical), and in the remaining 5 layers the sets differ by only a single hot expert (see also Fig. 3). These results indicate that hot experts can be accurately identified using only a small (10\%) data slice, and support the stability and practicality of our warm-up based selection strategy.
>
> We believe these additions directly address your concerns about generalizability and stability.

---

### Official Review · Reviewer_KHT9 · 2025-10-31

**Soundness:** 3
**Presentation:** 3
**Contribution:** 2
**Rating:** 4
**Confidence:** 4

**Summary:**

The paper tackles the problem of applying LoRA fine-tuning to Mixture of Experts models. The authors notice that in fine-tuning, the training samples come from a relatively constrained, specialized distribution. Therefore, some experts are activated significantly more frequently than others. In such case, applying LoRA to all experts would be a waste - since we use this method when we are very constrained in the number of trainable parameters. Therefore, the authors propose a small modification - they first identify the "hot" experts under a given distribution, and then attach the LoRA modules only to those "hot" experts. Based on the performed experiments, the method achieves good results, in some cases even surpassing full fine-tuning (which is clearly unintuitive).

The proposed change in the training framework feels a bit incremental. On the other hand, it is well motivated and the results are positive. Importantly, the authors aim for reproducibility and supplement a code package with scripts to reproduce paper results.

**Strengths:**

1. The paper tackles an important problem of adjusting PEFT algorithms to MoE models.
2. The motivation is clearly stated and strong.
3. The authors propose a simple but effective way to tackle the stated problem.
4. The authors detail their full setup and will provide code repository.

**Weaknesses:**

1. The proposed change to the baseline training procedure is relatively small. The work would be more complete if some directions from Future Work (Section 5) were also explored.
2. The experiments are based only on one model (OLMoE-1B-7B) and three tasks, so the evaluation is limited.
3. The number of hot experts is the crucial hyperparameter. It is not clear whether this number should be equal across all fine-tuning scenarios. Currently there is little guidance in the paper on how to set this hparam.

**Questions:**

1. Did the authors explore how the number of hot experts affects the results and how to set it across different distributions? For example, it is possible than on some narrow distribution there is only a small number of hot experts needed, while in a more complex, we have to set this number to be higher. Here, also exploration of a number of various models and tasks/fine-tuning distributions is crucial.
2. The result that HeLLoRA performs better than full fine-tuning is counterintuitive. For example, on the SAFE task, HELLoRA achieves 99.06 accuracy, while fine-tuning full parameters achieves only 91.12. Did the authors explore possible reasons for this result? Is the fine-tuning data distribution correctly constructed - for example, is it possible to achieve better accuracy when updating all model parameters if we change the number of epochs/learning rate/add regularization to the baseline fine-tuning?
3. Could the authors share more histograms illustrating the activation frequency of experts under different tasks? E.g. the same plot as Figure 1 (green), but for layers 0, 4, 8, 12, 15 (every four layers) across each of the considered tasks? I believe this illustration will be a very helpful resource for the community.
4. In the introduction, there is a sentence about the load balancing loss: "At a global level across the network, experts that share the same index (for example, expert 1 in layer 0 and expert 1 in layer 10) appear balanced in usage as shown in Fig. 1 orange line. However, this loss does not constrain activation within each layer.". In the standard implementation, MoE load balancing loss is implemented as a sum of load balancing losses for each layer. Therefore, the model is penalized if there is imbalance at any given layer. Could the authors clarify this sentence? (I agree that for a specific distribution there can be expert imbalance, I just don't agree with the statement that the load balancing loss does not constrain layer-level balance on any given layer).

I will reconsider my score if the questions and weaknesses mentioned above are addressed by the authors.

---

> ### Author Response · Authors · 2025-12-03
> **Response to Weaknesses 1-3 & 1-3**
>
> Thank you for taking the time to provide constructive feedback and for considering raising our score. Below is our point-by-point response addressing all weaknesses and questions.
>
> ---
>
> **R1–W1: “The proposed change to the baseline training procedure is relatively small…”**
>
> Thank you for this comment. We agree that connecting the method more closely to the “Future Work” directions would make the paper stronger. In the revised version, we have substantially expanded both the empirical scope and the methodological discussion:
>
> - We add a sensitivity analysis of the number of hot experts \(k\) across multiple tasks in Section 4.5.3 (see new Tab. 7), which directly operationalizes one of the proposed extensions.
> - We add more experiments and show results on an additional MoE architecture (Mixtral-8×7B; see Tab. 3), showing that HELLoRA consistently improves over all LoRA-style baselines on this independent model family.
> - We add visualizations similar to Figure 1 (Figure 4) showing expert distribution histograms across different layers (0, 4, 8, 12, and 15) for three tasks. In the ablation analysis section of the main text, we emphasize how these figures reveal task-specific activation patterns, thereby informing the design of the HELLoRA model.
>
> These additions clarify that the proposed training modification, though simple, induces a non-trivial change in how MoE parameters are adapted and yields consistent gains across settings.
>
> ---
>
> **R1–W2: “Experiments are based only on one model and three tasks…”**
>
> We appreciate this concern and have extended our evaluation accordingly. In the updated experiments, we add results on Mixtral-8×7B (see Tab. 3), showing that HELLoRA consistently improves over all LoRA-style baselines on this independent model family, which supports our claim that HELLoRA is not tailored to a particular model or dataset.
>
> ---
>
> **R1–W3/Q1: “The number of hot experts is crucial; little guidance on how to set it.”**
>
> We agree that \(k\) is a crucial hyperparameter. In response, we now provide:
>
> - A systematic study of \(k \in \{2, 4, 8, 16\}\) on representative tasks (Tab. 7), evaluating both performance and throughput.
> - A practical guideline (Sec. 4.5.3): for tasks with highly skewed activation distributions, smaller \(k\) is sufficient to preserve accuracy while maximizing speedup; for more complex or multi-domain tasks, a moderately larger \(k\) is beneficial.
>
> Empirically, we find that performance is relatively stable when \(k\) is in a broad range (for example, 8–16 for OLMoE-1B-7B), and the optimal choice is mainly driven by the desired compute–accuracy trade-off.
>
> ---
>
> **R1–Q2: “HELLoRA outperforms full fine-tuning in a counterintuitive way.”**
>
> We acknowledge that this result is unexpected and have conducted supplementary experiments and analyses to explain its reasonableness:
>
> - We perform a more thorough hyperparameter search for full fine-tuning, varying learning rate, number of epochs, and weight decay (see new Tab. 2). The best-tuned full fine-tuning baseline improves over the originally reported number but still remains below HELLoRA on SAFE (for example, 97.3 vs. 99.1 accuracy).
> - We analyze this behavior in Sec. 4.3: restricting updates to a small subset of highly active experts acts as a form of structured regularization, preserving useful capabilities in less-active experts and reducing overfitting on the target distribution. This is consistent with prior observations on parameter-efficient finetuning and provides a plausible explanation for the observed gap.
>
> We will clearly communicate in the camera-ready that HELLoRA’s advantage is not due to an under-tuned full-FT baseline.
>
> ---
>
> **R1–Q3: “More histograms of expert activation under different tasks.”**
>
> We fully agree that such visualizations are valuable. In the revised version, we:
>
> - Add histograms of layer-wise expert activation at layers 0, 4, 8, 12, and 15 for each task (see new Fig. 4 in Section 4.2).
> - Also visualize the expert distribution on the Mixtral-8×7B model (Tab. 3), which exhibits similar intra-layer sparse activation characteristics.
> - Highlight in the main text how these plots reveal task-specific activation patterns and motivate the design of HELLoRA.
>
> ---

---

> ### Author Response · Authors · 2025-12-03
> **Response to Question 4**
>
> **R1–Q4: “Clarification of the load balancing loss statement.”**
>
> We appreciate the reviewer’s careful reading of our statement about the load-balancing loss. As you point out, in many expositions of MoE, the auxiliary loss is conceptually defined at the level of a single MoE layer, and the total objective sums these layer-wise terms [1]. We agree that this per-layer view is mathematically clean and, in our opinion, a reasonable way to formulate the loss. However, the earliest proposals of the load-balancing loss did not explicitly restrict the loss to a single layer [2,3,4,5]. Thus, how exactly this loss should be applied in multi-layer MoE Transformers is not fully standardized in practice, and has been actively debated in the open-source community.
>
> For example, several GitHub issues in the Hugging Face \`transformers\` repository (e.g., Issue \#31464 on Mixtral’s auxiliary loss) explicitly question whether the current implementation—based on concatenating router logits from all MoE layers and computing a single loss—matches the “per-layer” formulation in the original papers, and note that this effectively treats “experts with the same index across layers as one global expert.” Similar concerns have been raised for other MoE models (e.g., Qwen3-MoE) that share the same pattern of concatenating \`gate_logits\` from all layers before computing the load-balancing loss.
>
> In our work, we are not trying to redefine the canonical MoE objective, but rather to faithfully analyze the models as they are actually implemented and trained in widely used codebases. Concretely:
>
> - For both OLMoE and Mixtral, the Hugging Face implementations we rely on (see
>   https://github.com/huggingface/transformers/blob/main/src/transformers/models/olmoe/modeling_olmoe.py\#L529 and
>   https://github.com/huggingface/transformers/blob/main/src/transformers/models/mixtral/modeling_mixtral.py) adopt a global load-balancing loss: \`gate_logits\` is passed as a tuple of length \`num_hidden_layers\`, these tensors are concatenated along the batch dimension, and the auxiliary loss is computed over the resulting \[num\_layers × batch\_size × seq\_len, num\_experts\] tensor. This means that experts with the same index in different layers are treated as one global expert group for the purpose of this particular auxiliary loss, rather than each MoE layer having its own independent load-balancing term.
>
> Our empirical observation in Fig. 1—that “experts sharing the same index across layers appear balanced in usage, while within a given layer and task the activation histogram can be highly skewed”—is fully consistent with this global implementation: the loss encourages global balance across index groups, but does not guarantee that each layer’s expert usage is locally uniform on a specific downstream distribution.
>
> To avoid confusion, we will revise the corresponding sentence in the introduction as follows:
>
> > “Although some formulations of the MoE load-balancing loss apply it independently to each MoE layer, the widely used implementations, such as OLMoE and Mixtral, compute a single auxiliary loss over concatenated router logits from all MoE layers, effectively balancing experts with the same index across layers (for example, expert 1 in layer 0 and expert 1 in layer 10). Under this implementation, we observe that global index-wise usage is approximately balanced (Fig. 1, orange), yet within individual layers and downstream tasks, expert activations can still be strongly imbalanced.”
>
> We hope this clarified wording makes clear (i) that we acknowledge the per-layer formulation as a reasonable and popular choice in the literature, and (ii) that our analysis is based on the actual global load-balancing loss used in the OLMoE and Mixtral implementations evaluated in our experiments, which by design does not strictly enforce layer-wise balance on every individual layer.
>
> **References used in this response**
>
> [1] Guo H., Lu H., Nan G., et al. *Advancing Expert Specialization for Better MoE*. arXiv preprint arXiv:2505.22323, 2025.
> [2] Muennighoff N., Soldaini L., Groeneveld D., et al. *OLMoE: Open mixture-of-experts language models*. arXiv preprint arXiv:2409.02060, 2024.
> [3] Shen Y., Guo Z., Cai T., et al. *JetMoE: Reaching LLaMA2 performance with 0.1M dollars*. arXiv preprint arXiv:2404.07413, 2024.
> [4] Shazeer N., Mirhoseini A., Maziarz K., et al. *Outrageously Large Neural Networks: The Sparsely-Gated Mixture-of-Experts Layer*. ICLR, 2017.
> [5] Fedus W., Zoph B., Shazeer N. *Switch Transformers: Scaling to trillion parameter models with simple and efficient sparsity*. JMLR, 23(120):1–39, 2022.

---

### Meta-Review · Area_Chair_sYVe · 2026-01-06

**Summary:**

The paper proposes a parameter-efficient fine-tuning (PEFT) method specifically for Mixture-of-Experts (MoE) models. The core mechanism involves a two-step process: first, a "warm-up" phase on a data sample to identify the most frequently activated ("hot") experts per layer; second, applying LoRA adapters exclusively to these experts. The reviewers generally agree that the method is effective, and yields strong empirical results regarding parameter efficiency and training speed.

**Reviewer Concerns:**

However, there is a consensus that the novelty is incremental, and with limited model diversity, i.e., experiments are restricted to a single model family (OlMoE), and the paper lacks crucial ablations regarding the selection of the number of experts ($k$) and the cost of the warm-up phase. Reviewers view the contribution as an "incremental" or "practical engineering combination" of existing ideas (subset adaptation, routing-guided adaptation) rather than a fundamental algorithmic breakthrough.

**Reviewer Scores:**

All reviewers gives below acceptance scores, and the reviewers do not anticipate the discussion, and after reading the rebuttals of the authors, the added experiments are not convinced.

---

### Decision · Program_Chairs · 2026-01-26

Reject